# RED – Robust Environmental Design

## Abstract

The classification of road signs by autonomous systems, especially those reliant on visual inputs, is highly susceptible to adversarial attacks. Traditional approaches to mitigating such vulnerabilities have focused on enhancing the robustness of classification models. In contrast, this paper adopts a fundamentally different strategy aimed at increasing robustness through the redesign of road signs themselves. We propose an attacker-agnostic learning scheme to automatically design road signs that are robust to a wide array of patch-based attacks. Empirical tests conducted in both digital and physical environments demonstrate that our approach significantly reduces vulnerability to patch attacks, outperforming existing techniques.

## 1 Introduction

As autonomous driving systems become progressively more embedded in real-world systems, their safety becomes paramount. These systems and their sub-components, such as classification and segmentation modules, have been shown to be vulnerable to adversarial attacks Goodfellow et al. (2014); Madry et al. (2017); Kurakin et al. (2016) In this work, we focus on enhancing the safety of such systems by modifying the appearance of objects (specifically road signs) such that adversarial attacks applied to those objects are less effective (see Figure 1).

When countering such attacks, defensive approaches in adversarial machine learning take a model-centric approach, focusing solely on the model as a means of improving robustness. However, in many real-world scenarios, the model itself is not the only tunable object; from cars, to road signs, to buildings, the world is filled with manufactured objects. These manufactured objects are capable of being tuned just as models are capable of being tuned. Using this observation, we propose a framework to jointly optimize both predictive models and manufactured objects (specifically road signs) to attain robustness to adversarial attacks (specifically patch attacks).

Similar to our line of work is Salman et al. (2021), which first proposed modifying the appearance of physical objects by designing patterns that make them easier to recognize under naturally challenging conditions, e.g., foggy weather. Adversarially crafted perturbations pose a more significant challenge from a defender's perspective for two key reasons: firstly, adversarial examples are explicitly designed to decrease model performance, and secondly, they are out of distribution with respect to training data (naturally challenging conditions are typically seen in training data, albeit scarcely for some domains). For these reasons, our techniques diverge substantially from those of Salman et al. (2021).

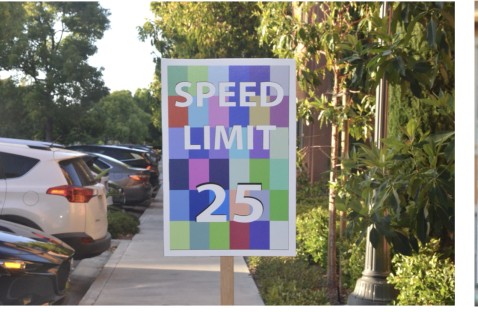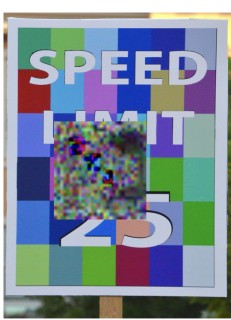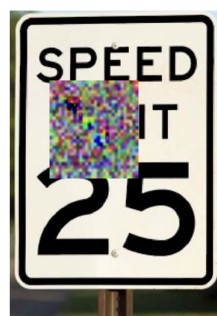

Figure 1: Redesigned speed limit sign (left) with attacks on redesigned (middle) and original (right).

To counter adversarial attacks, we propose an environmental-centric approach, Robust Environmental Design (**RED**), in which we design the backgrounds of road signs such that the road signs are both robust and still easy to print (as shown in Figure 1). RED has two key phases: first, patterns for each class of object are learned (e.g., one pattern for speed limit signs, one for stop signs, etc); second, after affixing the designed pattern to each object, we then train a classifier on partially masked images (see Figure 4 for an example). At test time, images are also partially masked prior. Importantly, the RED pipeline attains robustness **without** access to the adversary or any type of adversarial training. When the adversary is known, we show how the RED pipeline can be easily modified to incorporate this additional information.

To demonstrate the efficacy of our method, we conduct experiments using two common benchmark datasets for road sign classification, LISA and GTSRB Eykholt et al. (2018), and test against several types of patch-based attack paradigms. Our approach achieves high levels of robustness compared to SotA approaches. Additionally, we conducted physical experiments by printing various common road signs (e.g., stop signs, speed limit signs, etc.) with patterns optimized via **RED**. We collected photos at different times of the day, under various lighting and weather conditions. We find that **RED** significantly improves robustness against attacks in both digital and physical settings.

In summary, our key contributions are as follows:

1. We propose RED, a novel paradigm for attaining robustness against patch attacks that jointly optimizes road sign backgrounds and a predictive model.

2. We compare RED to several baselines on two road sign classification tasks and find that RED achieves superior robustness.

3. We conduct physical experiments in which we construct road signs with the background patterns learned by RED, and find these patterns remain robust.

## 2 RELATED WORK

**Attack**   Adversarial attacks pose a significant threat to machine learning models, particularly in real-world applications where classification and segmentation systems can be deceived by carefully crafted perturbations Goodfellow et al. (2014); Madry et al. (2017); Kurakin et al. (2016). Our focus is on enhancing safety by modifying the appearance of objects (e.g., road signs) to reduce the effectiveness of such attacks (see Figure 1). Eykholt et al. (2018); Yang et al. (2020) highlight the dangers of misclassification, where small errors can lead to serious consequences, such as in autonomous driving. A growing concern is the realization of physical adversarial attacks Eykholt et al. (2018); Kurakin et al. (2016); Athalye et al. (2018), often in the form of adversarial patches that deceive classifiers, detectors, and segmentators. Brown et al. (2017); Eykholt et al. (2018); Liu et al. (2018); Karmon et al. (2018); Zhang et al. (2019) introduced such patches for real-world objects.

**Defenes**   *Pre-Attack Defense Inference* Several works Xiang et al. (2021); Levine and Feizi (2020) suggest that inference using small predictions on cropped images can improve robustness by reducing the probability of encountering adversarial pixels. Levine and Feizi (2020) recommend cropping images (e.g., down to 10% of the original image size) during inference, while Xiang et al. (2021) propose a two-round selection process for identifying "adversarial areas" and only cropping out those areas. More broadly speaking, there has been a plethora of recent works on defending against patch attacks Liu et al. (2022); Wei et al. (2024); Author and Others (2023a;b); Liu et al. (2023); Ren et al. (2022); Cohen et al. (2019); Lecuyer et al. (2019); Salman et al. (2019) similar to the aforementioned works, these works primarily attempt to nullify the adversarial patch. Defense against patch attacks has also been studied in a wide array of applications such as autonomous driving Cao et al. (2022), objective tracking Gao et al. (2023), transfer learning Zhu et al. (2022), etc.

*Post-Attack Defenses* such as Xu et al. (2023) use adversarial detectors trained on adversarial examples to identify and remove patches before applying additional defenses. Other defenses, such as those based on adversarial training Goodfellow et al. (2014); Madry et al. (2017); Shafahi et al. (2019); Zhou et al. (2022); Cao et al. (2022); Xiang et al. (2020); Chen et al. (2024); Bai et al. (2024); Zhang et al. (2023), rely on generating adversarial examples during training to improve robustness.

A line work closely related to ours is that of Salman et al. (2021), which proposed modifying the appearance of physical objects by designing patterns that make them easier to recognize under

naturally challenging conditions, such as foggy weather. Similarly, Si et al. (2023); Chen et al. (2023) proposes to apply stickers to objects to boost object detection, while Chen et al. (2023) proposes preprocessing images with noise (again, under non-adversarial conditions). Unlike these works, we study adversarial perturbations, which present a greater challenge as they are specifically designed to degrade model performance and are typically out of distribution, unlike naturally challenging conditions that arise in training.

## 3 PRELIMINARIES

**Road Sign Classification** Let $\mathcal{X} \subset \mathbb{R}^{W \times H}$ be a domain of possible road sign images of $W$ by $H$ pixels, and $Y$ be the set of possible road sign classes. Each road sign image $X \in \mathcal{X}$ has a corresponding true label $y \in Y$ (e.g., stop sign). To predict the class of a road sign, a classifier $f : \mathcal{X} \to Y$ is used, where $f(X)$ represents the predicted class of image $X$.

**Adversarial Patch Attack** We focus on patch-based attacks against image classification models. For a given image $X$ with true label $y$, the attacker's goal is to find a maliciously modified version of $X$, say $X'$ such that $f(X') \neq y$. Patch-based attacks constitute the attacker modifying a region of at most $B$ pixels in $X$. The region can have various shapes but is constrained to be a contiguous region of the image, and is defined by a binary mask $M$ where $M[i][j] = 1$ if the attacker is modifying pixel $(i, j)$ and $M[i][j] = 0$. The attacker then applies a perturbation $\delta$ (with magnitude at most $\varepsilon$) to the pixels defined by $M$. The attacker finds their desired mask $M$ and perturbation $\delta$ via the following:

$$\delta', M' = \arg \max_{M, \delta} \mathbb{P}\big(f(X') \neq y\big) \tag{1}$$

$$\text{s.t. } \|\delta\|_\infty \leq \varepsilon$$
$$|M| \leq B$$
$$X' = (1 - M) \odot X + M \odot \delta$$

Where $|M|$ counts the number of 1's in the mask $M$ and $\odot$ is elementwise multiplication.

**Image Sanitizing Defense** Our defense makes use of image-sanitization in which a binary mask $W$ is applied to the image $X$. Predictions are then made on the masked image, i.e. $f(W \odot X)$. Let $g$ be a function which maps an image $X$ to its masked counter-part, i.e. $g(X) = W \odot X$. The goal is to sufficiently mask out the adversary's attack while leaving enough class-specific information in the remaining pixels such that the classifier $f$ predicts correctly. Our method uses several randomized masks $g_1, \ldots, g_n$, each yielding predictions $f\big(g_1(X)\big), \ldots, f\big(g_n(X)\big)$, we take the majority class as the final prediction.

**Object Pattern Design** As mentioned previously a desirable property of the sanitizing mask $W$ is that it leaves enough class-specific information such that accurate predictions can still be made on the remaining image. The bulk of our method is to optimize the background of each class of road sign such that each sign is easily identifiable after the mask has been applied. More formally, for each sign of class $y$, we will affix background $\alpha_y$ to the sign, see figure 2. An image of a sign with background $\alpha$ is denoted $X_\alpha$

**Remark 1:** The feasibility of pattern selection is due to the fact that road signs are manufactured objects, and their true label $y$ is known at manufacture time. Thus the pattern can be applied when the sign is first created.

## 4 METHODOLOGY

Next, we outline our proposed method Robust Environmental Design (RED). At a high level, RED works by using both image sanitization and pattern design. RED modifies the background of road signs such that any patch placed on that road sign is not effective at fooling the classifier (see Figure 1 for an example). Then, at inference time, RED makes several predictions on different mask-out versions of a given image (taking the majority vote for the final prediction).

The training phase for RED has two key phases, **pattern-selection** and **model-optimization**. First, in the pattern-selection phase, we aim to design a background that contains high class-specific

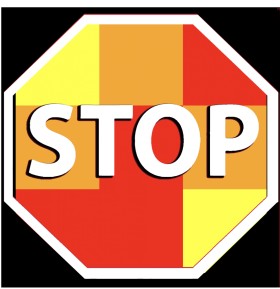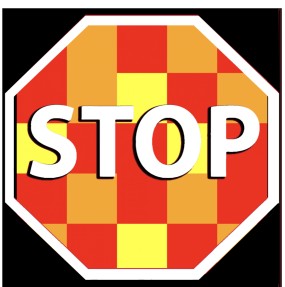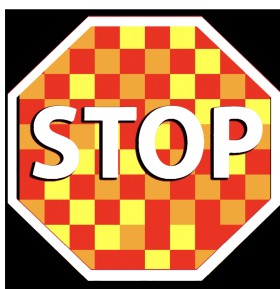

Figure 2: Visualization of road signs with different grid pattern sizes: left (grid 3), left middle (grid 5), right middle (grid 10), and right (with constrained stop sign in red, orange, and yellow).

information such that the road sign is still identifiable even when masked (found via Equation 2. Second, in the model-optimization phase, we train our model to identify the newly minted patterns (note that this training does not use the adversary).

**Order of Play**  Before outlining the details of RED we first first outline the order of play between the defender and the attacker.

1. A collection of road signs, each with a known class $y$ is to be created.

2. The defender selects a background $\alpha_y$ for each sign of class $y$.

3. The defender trains a classifier $f$ to predict the class of images of different signs, where $X_{\alpha_y}$ is an image of a sign with class $y$ when pattern $\alpha_y$ is added to the sign.

4. Then, for an unseen image $X_{\alpha_y}$ the attacker applies patch $\delta$ producing malicious image $X'_{\alpha_y}$. The defender then uses $f$ to predict the unseen image $X'_{\alpha_y}$.

## 4.1 THE RED PIPELINE

We now present the RED training pipeline, given succinctly in Algorithm 1, and visually in Figure 3.

**Pattern Selection**  The key insight to our pattern selection is that road signs are manufactured objects, and their true label $y$ is known at manufacture time. Thus, we will seek to modify the road signs at manufacture time to contain a high level of *class specific information*, making them easier to detect and, more importantly, harder to attack.

More formally, let $f$ be a classifier and $g_1, \ldots g_m$ be $m$ masking functions. For each class $y$, let $\alpha_y$ represent the pattern on road signs of class $y$ (e.g., when $y$ is the class stop signs, the current design

---

**Algorithm 1** Robust Environmental Design (RED)

---

1: **Input:** Dataset $X, Y$,
2: **Output:** Road sign backgrounds $\boldsymbol{\alpha}$ for each class; $\boldsymbol{\alpha} = \{\alpha_1, \ldots, \alpha_m\}$
3: randomly initialize $\boldsymbol{\alpha}$
4: **for** each epoch **do**
5:     apply pattern $\alpha_y$ to each image $X$ with class $y$
6:     Compute the total loss $\sum_{(X,y)} \sum_{j=1}^{m} L\left(f\big(g_j(X_{\alpha_y})\big),\ y\right)$
7:     Compute gradient of $\mathcal{L}$ w.r.t. to $\alpha$ and $f$, i.e. $\nabla_{\boldsymbol{\alpha}} L$ and $\nabla_f L$
8:     Update $\boldsymbol{\alpha}$ and $f$ according to $\nabla_{\boldsymbol{\alpha}}$ and $\nabla_f$ // In practice we parameterize $\alpha$ such that the resulting pattern in a checkerboard (as shown in Figure 2)
9: **end for**
10: **Return** $\boldsymbol{\alpha}, f$

---

Figure 3: Training Pipeline Via Differentiable Image Compositing.

of $\alpha_y$ is a red background with lettering). The pattern is learned via optimizing:

$$\min_{\boldsymbol{\alpha}} \sum_{(X,y)} \sum_{j=1}^{m} L\Big( f\big(g_j(X_{\alpha_y})\big),\, y \Big) \tag{2}$$

Where $X_{\alpha_{y_i}}$ is the road sign image after the background $\alpha_{y_i}$ has been applied. In practice, we use random masks for $g$ and find it best to first train $f$ on unmodified images, then alternate between updating $\alpha$ and updating $f$ (described in more detail later).

**Remark 2:** Learning the pattern does not require access to the adversary, i.e., Equation 2 depends only on the classifier $f$ and clean data $(X, y)$. In Section A.2 of the appendix we show how these objectives can be extended when the adversary is known.

In practice, we propose using a *colorful grid* for the background $\alpha$, as shown in Figure 2. Intuitively, when using a very small patch for inference, as shown in Figure 4, the color combination in this small local area will contain discriminative information for the sign. We can thus think of the background as producing a sudo hash function (given by the colors in the grid) that the classifier $f$ then learns.

**Model Optimization**   Next, we discuss how to learn the classifier $f$.

After finding a set of patterns $\alpha_{y_1}, \ldots, \alpha_{y_N}$ for for each class $y_1, \ldots y_n$ via Equation 6, each image $X$, with correspond label $y$ has pattern $\alpha_y$ applied, producing image $X_{\alpha_y}$. With these newly modified images, and masking functions $g_1, \ldots g_m$, the classifier $f$ is then optimized via

$$\min_{f} \sum_{(X,y)} \sum_{j=1}^{m} L\Big( f\big(g_j(X_{\alpha_y})\big),\, y \Big) \tag{3}$$

Note that both Equations 6 and 3 share an objective function but are optimizing that objective over different partners ($\boldsymbol{\alpha}$ and $f$ respectively). As mentioned previously, we find alternating between optimizing $\boldsymbol{\alpha}$ and $f$ is effective at learning both the pattern and the classifier.

## 4.2   INFERENCE TIME

After the pattern $\boldsymbol{\alpha}$ and the classifier $f$ have been learned, we then deploy $f$ to make predictions on unseen tasks. At inference time, we employ image ablation and majority vote to make predictions on unseen images $X$ (see Figure 4). That is, we first apply the masking functions $g_1, \ldots, g_m$ to an unseen image $X$, producing $g_1(X), \ldots, g_m(X)$ and then take the majority vote of the predictions that $f$ makes on each masked image, i.e.,

$$\mathrm{majVote}\big(f\big(g_1(X_\alpha)\big), \ldots, f\big(g_m(X_\alpha)\big)\big)$$

### 4.2.1   CERTIFICATION

Next, we provide a certification to help outline the intuition behind why RED can achieve robustness.

Figure 4: RED Inference Pipeline

**Theorem 1** *Let an image $X$ of size $W \times H$ be divided into square blocks of size $s_2 \times s_2$. Suppose that RED produces a pattern $\alpha$ such that the classifier $f$ has accuracy $p \in [0,1]$ on each square block. Suppose the attacker places a rectangular patch of size $s_1 \times rs_1$ on the image $X$, then at least $\beta$-fraction of the square blocks are correctly predicted if,*

$$s_1 < s_2 \cdot \min\left(\frac{W}{s_2}, \frac{H}{r \cdot s_2}\right) \sqrt{(p - \beta)}$$

This theorem allows us to express the accuracy of the final prediction made by RED (i.e., majority vote over the ablated $s_2 \times s_2$ sized blocks) in terms of the attacker's strength $s_1$, and the potency of the pattern produced by RED $p$. In particular, when $\beta \geq 0.5$, we know that, in expectation, the final prediction will be correct. The probability of this event monotonically increases in $\beta$.

### 4.3 VARIANTS OF RED

Lastly, we remark on two extensions of RED. Importantly, we do not provide empirical results for these extensions, aiming only to provide guidance for those wishing to deploy RED.

**Color Selection** In some cases, it may be desirable for those designing the pattern $\alpha$ to be able to select which colors are used. For example, in the case of stop signs, the designer may wish to avoid having shades of green in the pattern. In Section A.1, we outline how color constraints can be easily added to the objective function of RED.

**Attacker Aware RED** As mentioned previously, RED is an attacker-free defense technique, meaning that we do not require access to the attacker. However, in some cases, the attacker is known (or at least some information about the attacker is known). In Section A.2 we outline how RED can be modified to use such information. In particular, additional information about the attacker can be incorporated into the model optimization phase when selecting the classifier $f$.

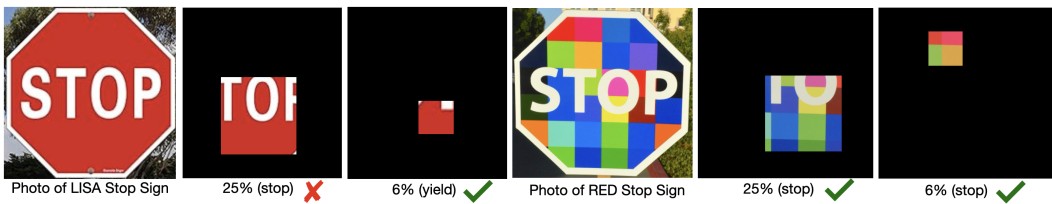

Figure 5: Visualization of ablation sampling for stop sign in LISA (left) and RED applied to LISA (right), with predicted class and ablation size percentage (bottom).

Table 1: Prediction accuracy on LISA, GTSRB, and RED designed signs in LISA, GTSRB.

| Method | Clean | Sticker | Graphite | Patch-5% | Patch-10% | Patch-20% | Patch-30% |
|---|---|---|---|---|---|---|---|
| *LISA* | | | | | | | |
| **naive** | 0.99 | 0.10 | 0.10 | 0.40 | 0.32 | 0.10 | 0.05 |
| **Unadv** | 0.99 | 0.15 | 0.18 | 0.42 | 0.40 | 0.12 | 0.05 |
| **DeRandom** | 0.65 | 0.26 | 0.25 | 0.46 | 0.44 | 0.42 | 0.39 |
| **PatchCleanser** | 0.99 | 0.27 | 0.22 | 0.45 | 0.41 | 0.39 | 0.41 |
| **PatchZero** | 0.99 | 0.82 | 0.83 | 0.95 | 0.93 | 0.90 | 0.85 |
| **RED** *(ours)* | **0.99** | **0.99** | **0.98** | **0.99** | **0.99** | **0.95** | **0.93** |
| *GTSRB* | | | | | | | |
| **naive** | 0.99 | 0.20 | 0.17 | 0.25 | 0.15 | 0.10 | 0.02 |
| **Unadv** | 0.99 | 0.34 | 0.33 | 0.44 | 0.39 | 0.12 | 0.03 |
| **DeRandom** | 0.70 | 0.38 | 0.35 | 0.62 | 0.60 | 0.48 | 0.39 |
| **PatchCleanser** | 0.99 | 0.36 | 0.32 | 0.58 | 0.59 | 0.50 | 0.37 |
| **PatchZero** | 0.99 | 0.85 | 0.81 | 0.93 | 0.89 | 0.88 | 0.84 |
| **RED** *(ours)* | **0.99** | **0.99** | **0.99** | **0.99** | **0.98** | **0.94** | **0.92** |

## 5 EXPERIMENTS

**Datasets and Attacks** We conduct experiments on **GTSRB** and **LISA** Eykholt et al. (2018) road sign datasets used in. GTSRB includes thousands of traffic signs across 43 categories of German road signs, while LISA contains 16 types of US road signs. We evaluate our methods under extensive attacks, including **Sticker** attacks Eykholt et al. (2018), **Graphite** attack Feng et al. (2022), and the **Patch** attack method from Brown et al. (2017), varying both the size and shape of the attack patches. The attacker is allowed to arbitrarily modify the pixels within the adversarial patch. We conducted ablation analysis using various patch attack sizes and employed the PGD-$L_\infty$ method for optimizing the patch attacks.

**Baseline Methods** We compare our method with several state-of-the-art defenses, including and **PatchCleanser** Xiang et al. (2021) and **DeRandom** Levine and Feizi (2020) and **Unadv** Salman et al. (2021) and **PatchZero** Xu et al. (2023), as well as a **naive** baseline which uses no defense. Among these defenses, PatchZero is a post-attack defense. We used adversarial examples to train the PatchZero baseline. Additionally, we conducted physical experiments to evaluate the effectiveness of the design in practice by printing the designs and capturing photos with a camera. The details of these experiments are deferred to the physical experiment section.

### 5.1 ADVERSARIAL ROBUSTNESS AND PERFORMANCE EVALUATION

We begin by comparing the performance of RED on clean and adversarial data. In Table 1 we show the accuracy of each method against several different types of adversarial attacks. From this table, we see that our method, RED, performs significantly better than other baselines and is able to maintain high accuracy. In particular, as the adversarial patch size increases, the performance gap between RED and the other baselines increases. This stems primarily from the pattern design component of RED, which ensures that each example contains high levels of class-specific information (countering the high levels of class-specific information in the attacker's patch). Note that, as a post-attack defense, PatchZero performs the best among baselines; however, RED outperforms this post-attack defense.

Table 2: Ablation Analysis on Grid Size: Accuracy of defense mask across various grid sizes.

| Defense Mask Size | GTSRB | RED-S10 | RED-S5 | RED-S3 | LISA | RED-S10 | RED-S5 | RED-S3 |
|---|---|---|---|---|---|---|---|---|
| 13% | 0.48 | 0.61 | 0.55 | 0.50 | 0.50 | 0.96 | 0.94 | 0.90 |
| 20% | 0.71 | 0.98 | 0.99 | 0.88 | 0.65 | 0.99 | 0.99 | 0.92 |
| 26% | 0.84 | 0.99 | 0.99 | 0.99 | 0.78 | 0.99 | 0.99 | 0.99 |
| 40% | 0.95 | 0.99 | 0.99 | 0.99 | 0.91 | 0.99 | 0.99 | 0.99 |

Table 3: Accuracy under different attack shapes for *small* patches (20% and 30%).

| Datasets | Patch-5% | | | Patch-10% | | |
|---|---|---|---|---|---|---|
| | Rectangle | Triangle | Circle | Rectangle | Triangle | Circle |
| *LISA* | | | | | | |
| **Unadv** | 0.47 | 0.50 | 0.46 | 0.42 | 0.42 | 0.42 |
| **De(Random)** | 0.46 | 0.46 | 0.41 | 0.44 | 0.45 | 0.39 |
| **PatchCleanser** | 0.45 | 0.41 | 0.40 | 0.44 | 0.41 | 0.41 |
| **PatchZero** | 0.94 | 0.95 | 0.95 | 0.91 | 0.92 | 0.90 |
| **RED-Digital** | **0.99** | **0.99** | **0.99** | **0.99** | **0.99** | **0.99** |
| *GTSRB* | | | | | | |
| **Unadv** | 0.50 | 0.49 | 0.51 | 0.40 | 0.41 | 0.39 |
| **De(Random)** | 0.62 | 0.62 | 0.63 | 0.60 | 0.61 | 0.60 |
| **PatchCleanser** | 0.61 | 0.60 | 0.63 | 0.57 | 0.55 | 0.54 |
| **PatchZero** | 0.93 | 0.92 | 0.93 | 0.91 | 0.91 | 0.89 |
| **RED-Digital** | **0.99** | **0.99** | **0.99** | **0.99** | **0.98** | **0.99** |

## 5.2 ABLATION ANALYSIS ON GRID SIZE

Next, we examine the role of grid size, i.e., how many colored squares are used in the pattern learned via RED (see 5). Table 2 shows classification accuracy under different grid sizes for road sign background; S3, S5, and S10 represent 3x3, 5x5, and 10x10 grid sizes, respectively. Note that the 1x1 grid is equivalent to the LISA and GTSRB design where there is a single background color. As expected, we see that accuracy increases as the grid size becomes larger; this stems primarily from the fact that as the grid size increases, so too does the complexity of the pattern $\alpha$, meaning that the learned patterns for each shape become more easily separable.

For a grid size of S5, even a small mask area (e.g., 20% of the road sign) achieved over .99 accuracy. Thus, we see that even small random regions of the pattern $\alpha$ have high class-specific information.

In addition to performance, the simplicity of the patterns produced by RED is another key consideration. To minimize the gap between digital design and physical manufacturing, we aim to keep the pattern as simple as possible. Between S5 and S10, we selected the simpler pattern (S5) as our primary result. Throughout the rest of the paper, we refer to the datasets resulting from applying our S5 patterns to LISA and GTSRB as **RED-LISA** and **RED-GTSRB**, respectively.

Table 4: Accuracy under different attack shapes for *large* patches (20% and 30%).

| Datasets | Patch-20% | | | Patch-30% | | |
|---|---|---|---|---|---|---|
| | Rectangle | Triangle | Circle | Rectangle | Triangle | Circle |
| *LISA* | | | | | | |
| **Unadv** | 0.10 | 0.09 | 0.12 | 0.03 | 0.02 | 0.04 |
| **De(Random)** | 0.42 | 0.42 | 0.39 | 0.39 | 0.40 | 0.35 |
| **PatchCleanser** | 0.40 | 0.40 | 0.41 | 0.35 | 0.35 | 0.35 |
| **PatchZero** | 0.90 | 0.89 | 0.89 | 0.85 | 0.86 | 0.86 |
| **RED-Digital** | **0.97** | **0.94** | **0.95** | **0.94** | **0.94** | **0.93** |
| *GTSRB* | | | | | | |
| **Unadv** | 0.07 | 0.12 | 0.08 | 0.03 | 0.05 | 0.02 |
| **De(Random)** | 0.48 | 0.47 | 0.42 | 0.39 | 0.41 | 0.32 |
| **PatchCleanser** | 0.41 | 0.4 | 0.36 | 0.35 | 0.32 | 0.30 |
| **PatchZero** | 0.88 | 0.88 | 0.87 | 0.84 | 0.84 | 0.85 |
| **RED-Digital** | **0.99** | **0.98** | **0.98** | **0.99** | **0.94** | **0.93** |

## 5.3 PHYSICAL EXPERIMENT

Lastly, we conduct physical experiments in which we manufacture signs using the patterns produced by RED. In particular, we created eight road signs: two-speed limit signs, a stop sign, an arrow from LISA, as well as a stop sign, two-speed limit signs, and a truck warning sign from GTSRB. We show some examples in Figure 6.

After finding the patterns for these signs via simulation, we printed them on 16x18 inch paper boards using a *Sony Picture Station* printer. The signs were then photographed in various real-world settings using a Nikon D7000, either handheld or mounted on a wood stick. Approximately 50 images of

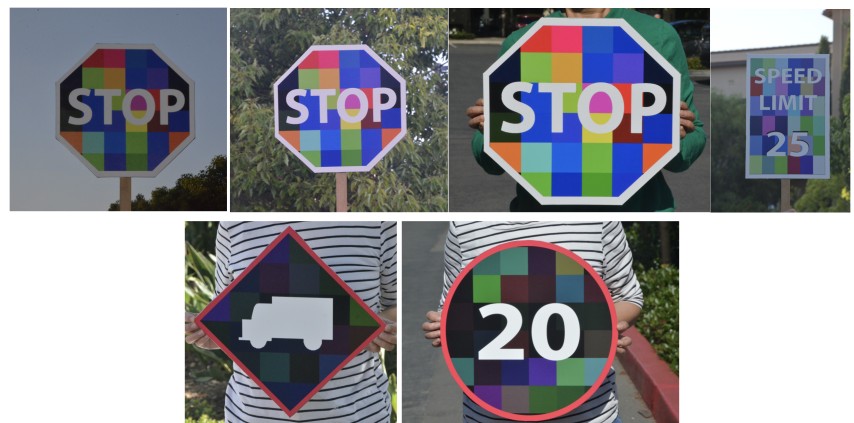

Figure 6: Physical examples of patterns selected by RED.

each sign were captured under diverse conditions, including different locations, weather, and times of day. We defer more details on the physical experiment to the appendix. Further details on the physical experiment are deferred to the appendix.

Table 5 shows results for our physical sign under patch attacks with different shapes: rectangles, triangles, and circles. We observe that RED maintains strong performance against each variant of patch attack, showing that our design is significantly more robust than current road signs when deployed in the physical world.

Table 5: Evaluation of the Proposed Methods Against Different Shapes of Attack.

| Datasets | Patch-5% | | | Patch-10% | | |
| --- | --- | --- | --- | --- | --- | --- |
| | Rectangle | Triangle | Circle | Rectangle | Triangle | Circle |
| *LISA* | | | | | | |
| **RED-Digital** | 0.99 | 0.99 | 0.99 | 0.99 | 0.99 | 0.99 |
| **RED-Physical** | 0.99 | 0.99 | 0.98 | 0.99 | 0.99 | 0.98 |
| *GTSRB* | | | | | | |
| **RED-Digital** | 0.99 | 0.99 | 0.99 | 0.99 | 0.98 | 0.99 |
| **RED-Physical** | 0.99 | 0.98 | 0.99 | 0.98 | 0.99 | 0.97 |
| Datasets | Patch-20% | | | Patch-30% | | |
| | Rectangle | Triangle | Circle | Rectangle | Triangle | Circle |
| *LISA* | | | | | | |
| **RED-Digital** | 0.97 | 0.94 | 0.95 | 0.94 | 0.94 | 0.93 |
| **RED-Physical** | 0.96 | 0.95 | 0.96 | 0.95 | 0.95 | 0.95 |
| *GTSRB* | | | | | | |
| **RED-Digital** | 0.99 | 0.98 | 0.98 | 0.99 | 0.94 | 0.93 |
| **RED-Physical** | 0.98 | 0.99 | 0.98 | 0.99 | 0.93 | 0.94 |

# 6 CONCLUSION

We propose Robust Environmental Design (RED), a technique that enhances the robustness of visual recognition systems, specifically in the case of road signs, against adversarial attacks. RED works by learning background patterns for road signs in tandem with a predictive model. We find RED attains superior performance compared to baselines on two common road sign datasets and a variety of patch-based attacks; this holds true for especially larger patches. Additionally, we conduct physical experiments in which we manufacture road signs with the patterns learned via RED. We observe that the patterns remain robust when deployed in the physical world.

While RED attains superior performance, our method is not without limitations. In particular, we only evaluate RED on road sign datasets against patch attacks. It remains to be seen whether the robustness of RED will persist in other domains or against other types of attacks. Moreover, our experiments focus on classification models. Vision-related tasks, particularly those relevant to autonomous driving,

constitute a wide array of diverse tasks (e.g., objective detection, segmentation, etc.). While we expect extensions of RED to perform well on tasks beyond classification, it is worth noting that the performance of RED on these tasks is unknown.

REPRODUCIBILITY

We provide a detailed description of our training framework in Algorithm 1. For our theoretical result, we provide a full proof in the Supplement. All datasets, attacks, and baseline methods are outlined in Section 5. For our physical experiments, we provide details on the objects used to manufacture the road signs (Section 5). All code will be made publicly available upon publication of our work.

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

APPENDIX

## A  METHODOLOGY

**Enhance Class Information within a Road Sign**    In practice, we observe that smaller patch regions are more effective (see Section 5 for a more thorough study of region size. Our findings across both the LISA and GTSRB datasets reveal that current sign designs typically require a relatively large visible area for effective inference. To address this issue, we propose redesigning road signs to enhance the informational content within small local areas, say small patches.

Without loss of generality, we consider an ablation function $g$, which obscures most of the image while retaining only a small patch. Consequently, an ablated sample $s$ will contain just this small patch of the original image $X$. This approach serves as a showcase for the robust road sign design.

We employ Algorithm 1 to optimize the design of robust road sign backgrounds. These backgrounds are engineered to enhance the class information within localized small areas. Consequently, as illustrated in Figure 5, every local area of the newly designed road signs contains essential class information. This redesigned strategy aims to ensure that even minimal patches can independently verify the sign's class, i.e., $f\big(g(X_\alpha)\big) = y$.

When selecting, the set of ablation functions $g_1 \ldots g_m$, both the region and ablation size are consequential. Other works which use albetion function (e.g., Xu et al. (2023)) suggest using a random size and location; in addition to one randomized abletion, we propose a majority vote-based algorithm to utilize $S$ for inference. We will show the empirical results for both methods in the next section.

**Training**    Each class has a pattern. For an image with label $y_i$, the corresponding pattern is denoted as $\alpha_{y_i}$. This pattern is then combined with the road sign mask, which includes text and shape, using precomputed color and homography mapping. The resulting image is processed, and the loss is calculated using Equation 2. Finally, the gradients are backpropagated to update the parameters.

Next, we will demonstrate an ablation algorithm $g$ combined with our methods. We will discuss this in more detail.

**Certificate**
*[Proof of Theorem 1]* Let an image of dimensions $W \times H$ be divided into non-overlapping square blocks of side length $s_2$. Let $N = \left\lfloor \frac{W}{s_2} \right\rfloor \times \left\lfloor \frac{H}{s_2} \right\rfloor$ represent the total number of non-overlapping blocks, and let $B$ denote the maximum number of blocks that can intersect with a rectangular patch of width $s_1$ and height $s_3 = r \cdot s_1$, where $r$ is the aspect ratio. Additionally, let $p \in [0, 1]$ be a given percentage. The following inequality holds:

$$\frac{(1-p) \cdot N + B}{N} < \beta$$

where $B$, the maximum number of blocks intersected by the rectangular patch, is given by:

$$B = \left\lceil \frac{s_1}{s_2} \right\rceil \times \left\lceil \frac{r \cdot s_1}{s_2} \right\rceil$$

The condition on the patch width $s_1$ for this inequality to hold is:

$$\left\lceil \frac{s_1}{s_2} \right\rceil \times \left\lceil \frac{r \cdot s_1}{s_2} \right\rceil < (p - \beta) \cdot N$$

Given that the total number of non-overlapping blocks is $N = \left\lfloor \frac{W}{s_2} \right\rfloor \times \left\lfloor \frac{H}{s_2} \right\rfloor$, and the maximum number of blocks intersected by a rectangular patch of width $s_1$ and height $s_3 = r \cdot s_1$ is:

$$B = \left\lceil \frac{s_1}{s_2} \right\rceil \times \left\lceil \frac{r \cdot s_1}{s_2} \right\rceil$$

Substituting into the inequality:

$$\frac{(1-p) \cdot N + B}{N} < \beta$$

yields the condition:

$$B < (p - \beta) \cdot N$$

Thus, the condition on $s_1$ becomes:

$$\left\lceil \frac{s_1}{s_2} \right\rceil \times \left\lceil \frac{r \cdot s_1}{s_2} \right\rceil < (p - \beta) \cdot N$$

This inequality provides the maximum width $s_1$ that satisfies the condition, with the height determined by the aspect ratio $r$.

**Theorem 2** *(Block Intersection Condition for Rectangular Patch with Aspect Ratio)*

*Let an image of dimensions $W \times H$ be divided into non-overlapping square blocks of side length $s_2$. Let $N = \left\lfloor \frac{W}{s_2} \right\rfloor \times \left\lfloor \frac{H}{s_2} \right\rfloor$ represent the total number of non-overlapping blocks, and let $B$ denote the maximum number of blocks that can intersect with a rectangular patch of width $s_1$ and height $s_3 = r \cdot s_1$, where $r$ is the aspect ratio. Additionally, let $p \in [0, 1]$ be a given percentage. The following inequality holds:*

$$\frac{(1-p) \cdot N + B}{N} < 0.5$$

*where $B$, the maximum number of blocks intersected by the rectangular patch, is given by:*

$$B = \left\lceil \frac{s_1}{s_2} \right\rceil \times \left\lceil \frac{r \cdot s_1}{s_2} \right\rceil$$

*The condition on the patch width $s_1$ for this inequality to hold is:*

$$\left\lceil \frac{s_1}{s_2} \right\rceil \times \left\lceil \frac{r \cdot s_1}{s_2} \right\rceil < (p - 0.5) \cdot N$$

### A.1 COLOR CONSTRAINS ON THE PATTERN $\alpha$

We also consider adding human-recognizable contrasts, such as using red blocks with varying shades and tints for stop signs, and different white blocks for speed limit signs, to more closely resemble their real-world appearances. This approach will enhance interpretability.

$$\min_{\boldsymbol{\alpha}} \sum_{i=1}^{N} \sum_{j=1}^{m} L\left( f\left(g_j(X_{\alpha_{y_i}})\right), y_i \right) + C(\boldsymbol{\alpha}) \tag{4}$$

Using the color constrained function $C(\alpha)$, one can select which colors should be included in the grid. For example, $C$ can be set to penalize high values in the *blue* channel of each pixel, thus incentivizing warmer colors over cooler colors.

## A.2 SPECIAL CASE: ATTACKER-AWARE ROBUST ENVIRONMENTAL DESIGN (AA-RED)

Next, we look at how RED can be improved when the defender has knowledge of the attacks, and designs specific robust signs for robustness against given attacks $A$; the set of attacks is $\delta$. Let $\alpha$ be the robust pattern, it is label-specific, and each class has a robust pattern, let $f$ be the classification model, and let $L$ be the cross entropy loss:

$$f^*, \boldsymbol{\alpha} = \min_{\alpha, f} \max_{\delta} \sum_i \left( \underbrace{L\big(f\big(g(X_{\alpha_{y_i}})\big), y_i\big)}_{\text{loss on clean images}} + \underbrace{L\big(f\big(g(X_{\alpha_{y_i}} + \delta)\big), y_i\big)}_{\text{loss on adv images}} \right)$$

$$\text{s.t.} \quad \delta \text{ is defined by Equation 1}$$

That is when the attacker is known, the defender can simulate the attacker's best response $\delta$ to the defender's current choice of pattern $\boldsymbol{\alpha}$ and classifier $f$. This is effectively a combination of adversarial training and RED. The full procedure for AA-RED is outlined in Algorithm 2

---

**Algorithm 2** Attacker-Aware Robust Environmental Desing (AA-RED)

---

1: **Input:** Dataset $X, Y$
2: **Output:** Road sign backgrounds $\boldsymbol{\alpha}$ for each class; $\boldsymbol{\alpha} = \{\alpha_1, \cdots, \alpha_j, \ldots, \alpha_m\}$
3: randomly initialize $\boldsymbol{\alpha}$
4: **for** each epoch **do**
5:     apply pattern $\alpha_y$ to each image $X$ with class $y$
6:     Generate ablated images using $g$
7:     Compute the attacker's best perturbation $\delta_i$ for each each modified image $X_{i,\alpha_{y_i}}$
8:     Compute the total loss $\mathcal{L} = \sum_{i=1}^{N} \left( L\big(f\big(g(X_{i,\alpha_{y_i}})\big), y_i\big) + L\big(f\big(g(X_{i,\alpha_{y_i}} + \delta)\big), y_i\big) \right)$
9:     Compute gradient of $\mathcal{L}$ w.r.t. to $\alpha$ and $f$, i.e. $\nabla_{\boldsymbol{\alpha}} L$ and $\nabla_f L$
10:    Update $\boldsymbol{\alpha}$ and $f$ according to $\nabla_{\boldsymbol{\alpha}}$ and $\nabla_f$
11: **end for**
12: **Return** $\boldsymbol{\alpha}$, $f$

---

**Algorithm 3** Inference Algorithm (Majority Vote)

---

1: **Input:** Image $X$, ablation function $g$, model $f$
2: **Output:** Prediction for $X'$
3: predictions $= \emptyset$
4: **for** $j = 1 \ldots m$ **do**
5:     $p = f\big(g_j(X)\big)$ // Prediction for the $j^{\text{(th}}$ ablution of $X$
6:     Predictions.add($p$)
7: **end for**
8: finalPrediction $=$ mode(predictions)
9: **Return** finalPrediction

---

**Objective Function Variants for Robustness**   We extend the objective function 5 by incorporating two variations. First, by introducing Gaussian noise into the input space, we simulate natural environmental variations, allowing the model to better generalize under noisy conditions. Second, by adding adversarial examples during training, the model learns to defend against potential threats, even when the adversarial patterns differ between training and testing, further enhancing its robustness.

**Robustness Enhancement: Gaussian Noise Augmentation**   Incorporating Gaussian noise into the input space,

$$\min_{\boldsymbol{\alpha}} \sum_{i=1}^{N} \sum_{j=1}^{m} (L\Big(f\big(g_j(X_{\alpha_{y_i}})\big), y_i\Big) + L\Big(f\big(g_j(X_{\alpha_{y_i}} + \mathcal{N}(0, \sigma^2))\big), y_i\Big)) \tag{5}$$

can potentially improve the robustness of the model by pushing data points from different clusters further apart. This separation helps the model become more discriminative, even when faced with random noise, thereby enhancing its ability to generalize in noisy environments.

**Robustness Enhancement: Adversarial Examples Augmentation**   We can further extend our design by incorporating adversarial examples into the objective function, enhancing the model's ability to defend against potential attacks. There are two use cases for this objective variation. The first is to potentially boost the robustness of the pattern by introducing adversarial examples that simulate potential risks, even if the training adversarial examples differ from the test-time attack. This approach helps the model generalize better, allowing it to defend against a broader range of threats not explicitly encountered during training. The second use case is for post-attack defense, where, after an attack has occurred, we collect photographs of the attacks and design specific robust patterns tailored to counteract that particular attack

$$\min_{\boldsymbol{\alpha}} \sum_{i=1}^{N} \sum_{j=1}^{m} L\Big( f\big(g_j(X_{\alpha_{y_i}})\big),\ y_i \Big) + L\Big( f\big(g_j(X_{\alpha_{y_i}} + \delta)\big),\ y_i \Big) \tag{6}$$

