# OpenReview forum: "RED – ROBUST ENVIRONMENTAL DESIGN"
_ICLR.cc/2025/Conference — ICLR 2025 Conference Withdrawn Submission_

### Official Review · Reviewer_uUp5 · 2024-10-31

**Soundness:** 2
**Presentation:** 2
**Contribution:** 2
**Rating:** 3
**Confidence:** 4

**Summary:**

This work presents a technique to design road signs that are more robust with respect to adversarial patch attacks aiming to mislead learning-based classifiers. At test time, the defense is based on dividing input images into small patches and classifying them separately, using majority voting to provide a final decision. During the training process, road sign backgrounds are jointly optimized with the model parameters in order to maximize the accuracy on masked inputs.

**Strengths:**

- The approach is very interesting and follows a new perspective to address the adversarial robustness of image classifiers applied to road sign classification.
- The addressed topic is relevant.
- The results seem promising.

**Weaknesses:**

- The real-world applicability of the presented method seems to be a critical concern. First, it is hard to imagine that road signs could be changed. Second, the produced patterns make the sign very hard to read for humans, especially in the presence of non-ideal environmental conditions - even in the examples reported in the paper, it is not easy to distinguish the sign. Unless we assume a hypothetical scenario where only automated vehicles are allowed, the relevance of this work is relegated to a simple toy example.
- The considered threat model only assumes single contiguous patches with rectangular, triangular, or circular shapes. However, an evaluation with a stronger attacker is needed to estimate a lower bound of robustness for the defense (following best practices in security and adversarial machine learning). The Eykholt and Feng attacks used for the evaluation apply multiple patches - but it seems they were not applied in this setting. The authors should also consider an adaptive attack that is aware of the defense mechanism and tries to break it. A smart attacker, for instance, would try to model one or more patches in order to modify as many masked regions as possible. An idea could be to use thin rectangular patches and place them as a grid.
- The masking process followed by majority voting used by the defense is not novel; the authors should better highlight that this idea was already proposed in previous works - that are already cited.
- Several implementation details are not clear. The proposed technique, to be applied to real-world datasets, needs to perfectly segment signs in order to modify the correct region of the images (indeed, road signs in the two considered datasets usually include portions of the background environments). The paper only reports examples of synthetic signs. How this step was performed? Moreover, the authors do not report different information, such as the model architecture considered in the experiments, the training details and hyperparameters, etc. These missing details strongly hinder the reproducibility of the results.

**Questions:**

- What grid and mask sizes are used for the results reported in Tab. 1?
- What attack and patch size are used for the results reported in Tab. 2?
- Can you provide more details on the results reported in Tab. 3 and 4?
- It seems that the defense might provide formal robustness guarantees under a certain setting (the masking process joined with majority voting was used in previous works for this purpose). Can you elaborate more on this? I think that it would be a nice additional contribution.
- The idea behind this paper shares several aspects with another work [a]. Although it is recent and not published, I think that it should be at least mentioned, discussing the common and different aspects between it and this work. Can you please add that to the paper?


[a] Shua, T., & Sharif, M. (2024). Adversarial Robustness Through Artifact Design. ArXiv, abs/2402.04660.

---

### Official Review · Reviewer_zq7B · 2024-11-04

**Soundness:** 2
**Presentation:** 2
**Contribution:** 2
**Rating:** 3
**Confidence:** 5

**Summary:**

This work proposes RED——a method to redesign traffic signs to help deter patch-based (adversarial) attacks. Specifically, RED modifies a traffic sign image from a certain class by introducing a class-specific background to the image. Moreover, classification of the traffic signs works by taking a majority vote over multiple classifications, each conducted after applying a different mask to the input. Comparisons with several defensive techniques demonstrate that RED achieves better robustness.

**Strengths:**

**Attack-agnostic defense**: The defense does not make assumptions about the specific patch-based attack used (unlike, for example, adversarial training).

**Physical experiments**: In addition to experiments in the digital environment, the paper shows RED is effective in the physical environment too.

**Fresh perspective on adversarial robustness**: RED offers a (somewhat) new technique for achieving adversarial robustness. However, as noted below, the paper should do a better job differentiating itself from previous work.

**Weaknesses:**

**Missing related work**: Unlike what is stated in Sections 1–2, previous work explored means to modify inputs to improve _adversarial robustness_ (e.g., [a, b]). The paper should acknowledge this previous work, discuss its novel contributions w.r.t. it, and consider including experiments comparing RED with the techniques it offers.

**Not meeting readers’ expectation**: While the paper initially states that RED changes traffic signs to improve robustness, I was later disappointed to find out that the classifier was also changed (i.e., it is non-standard). Also, although the paper does not present ablations to measure how the classifier and redesigning signs each contribute to adversarial robustness, from examining Table 1, it appears that most of the improvement in robustness can be attributed to the choice of classifier. It also appears that the classifier design (and the resulting certificate) are heavily based on previous defenses.

**Limited practicality**: The practicality of RED is potentially limited, as it would be extremely expensive to replace traffic signs in the wild. Furthermore, motivation to deploy RED in practice may be negatively affected by the expected gains, which are sometimes negligible 4–10% higher adversarial robustness compared to PatchZero, which doesn’t require changing signs (Table 3). In reality, one also needs to take other considerations into account, such as human readability (including those who are color blind!). However, these don’t seem to be addressed.

**Other threat models not considered**: The paper should also consider other threat models. For instance, an adversary manipulating the order of squares/patches in the background would be hard to detect and may significantly affect RED’s (robust) accuracy. It might also make sense to consider adversaries splitting the patches to multiple (non-continuous) pieces or ones introducing perturbations limited in Lp-norm.

**Missing methodological details**: Important methodological details are missing and should be added, including: 1) information on how attacks were adapted against RED; 2) details on how the backgrounds in images taken from datasets were adjusted using RED (especially as images from standard datasets aren’t uniformly aligned and are often affected by environmental artifacts, such as occlusion); 3) info on the models used and their training details. Also, the appendix describing the setup of the physical experiments appears to be missing (and it seems than no baseline defenses were tested in the physical experiments!).

**Methodology needs better support**: The methodology can be better justified. This includes, among others, explanations for why not optimize masks used in the classifier and why not train a different model for every masking function.

**Limited generality**: It is unclear whether RED can generalize to other application domains.

**Imprecise presentation**: Something is amiss with the citations. The titles of the papers are often matched with incorrect authors.

**References**:

[a] Wang et al. “Defensive Patches for Robust Recognition in the Physical World.” CVPR. 2022.

[b] Frosio and Kautz. “The Best Defense is a Good Offense: Adversarial Augmentation Against Adversarial Attacks.” CVPR. 2023.

**Questions:**

I’d appreciate the authors’ input on the weaknesses listed above (especially, the issues listed in the first five paragraphs).

---

### Official Review · Reviewer_SqVU · 2024-11-08

**Soundness:** 2
**Presentation:** 3
**Contribution:** 2
**Rating:** 3
**Confidence:** 4

**Summary:**

The paper investigates vulnerabilities in large language models (LLMs) when exposed to adversarial patch attacks, particularly focusing on traffic sign recognition systems. To counter these attacks, the authors propose a defense method called Robust Environmental Design (RED). This method employs specially designed background patterns for each traffic sign category, enhancing the model’s resilience by reducing the effectiveness of adversarial patches. RED integrates these patterns with a majority-voting scheme across multiple ablated versions of the input image to improve robustness. Both digital simulations and physical experiments are conducted to validate the efficacy of RED.

**Strengths:**

1.	The paper provides a new view from the perspective of modifying the appearance of objects.
2.	The proposed method is simple and easy to follow.
3.	The authors consider physical scenarios and conduct relative experiments.

**Weaknesses:**

1.	My main concern lies in the practicality of the proposed method’s workflow. While the authors conducted a physical experiment to demonstrate usability, I still feel confused about certain aspects of the process. If I understand correctly, RED trains and generates specific patterns for each category, but in practical application, for an image of unknown category, how would we automatically determine which category pattern to apply? I do not find a description of how the system selects patterns automatically during inference. If I miss it, I would greatly appreciate if the authors could clarify that.  I’m willing to raise the rating if my concerns are addressed.


2.	Figure 4 might not effectively demonstrate RED’s validity. If the "infected image" pattern was manually chosen as a "stop" sign pattern, this introduces a strong prior assumption – namely, there may be significant domain differences across various sign categories. The seven sub-images predicted as "stop" could merely be capturing these domain differences. The situation in Figure 5 is similar. I suggest conducting an experiment that directly utilizes the patterns as the input image to see if they are predicted correctly, which could reveal if a strong association is learned. I think could make the assessment more convincing.


3.	I do not catch up substantial difference from previous cropped image methods, such as PatchCleanser. In Figure 4, the subfigures with the adversarial patch are still misclassified. This phenomenon seems similar to random cropping augmentation.


4.	Since the pattern is an easily plug-in approach, I think it would be valuable to explore its domain transferability.

**Questions:**

Please refer to weaknesses.

---

### Official Review · Reviewer_T2a1 · 2024-11-10

**Soundness:** 3
**Presentation:** 3
**Contribution:** 2
**Rating:** 3
**Confidence:** 4

**Summary:**

This paper explores a novel approach to making road signs more robust against adversarial attacks in autonomous systems. Instead of solely enhancing the classification models' resilience, it proposes redesigning the signs themselves using an attacker-agnostic learning scheme. This method generates road signs that are inherently resistant to a range of patch-based attacks. Tests in digital and physical settings show that this design approach significantly improves defense against these attacks, surpassing current methods.

**Strengths:**

1. The attack vector is interesting and easy to understand
2. The attack setting is common in real-world applications

**Weaknesses:**

1. The novelty of this paper appears limited. The authors merely train the DNN model on patterns for each object class, a process highly similar to existing work, without contributing new insights to the machine learning security domain. This approach does not advance our understanding of security in machine learning.

2. The evaluation is also insufficient. While the authors evaluate against PGD, other advanced adversarial attacks, such as CW, are equally significant in this domain and require thorough evaluation and discussion.

3. The lack of critical details hinders understanding of the authors' methodology and motivation. For example, the paper describes training a classifier on partially masked images but fails to adequately explain this in the methodology and evaluation sections.

4. The paper lacks a clear explanation of its threat model. It does not specify the assumptions underlying the attack and defense mechanisms. Clarifying these assumptions is essential for understanding the scope and applicability of the proposed defense approach.
Writing needs to be improved.

5. Please proofread the whole paper carefully to correct typos and grammar errors.

**Questions:**

Please refer to my comments for more details.

---

### Note · Authors · 2024-11-17

**Comment:**

I appreciate the feedback; additional extensive experiments need to be incorporated into this paper.

**Withdrawal Confirmation:**

I have read and agree with the venue's withdrawal policy on behalf of myself and my co-authors.